# Hyperexpression of CDRs and HWP1 genes negatively impacts on *Candida albicans* virulence

**Bruno Maras**[1]☉, **Anna Maggiore**[1]☉, **Giuseppina Mignogna**[1], **Maria D'Erme**[1], **Letizia Angiolella**[2]☉*

1 Dipartimento di Scienze Biochimiche "A. Rossi Fanelli", Sapienza Universita'di Roma, Rome, Italy,
2 Dipartimento di Sanita'Pubblica e Malattie Infettive, Sapienza Universita'di Roma, Rome, Italy

☉ These authors contributed equally to this work.
* letizia.angiolella@uniroma1.it

**Data Availability Statement:** All relevant data are within the paper.

**Funding:** LA, GM, BM. Grant number: RM 1181641E9150B3 Sapienza University of Rome

## Abstract

*C. albicans* is a commensal organism present in the human microbiome of more than 60% of the healthy population. Transition from commensalism to invasive candidiasis may occur after a local or a general failure of host's immune system. This transition to a more virulent phenotype may reside either on the capacity to form hyphae or on an acquired resistance to antifungal drugs. Indeed, overexpression of genes coding drug efflux pumps or adhesins, cell wall proteins facilitating the contact between the fungus and the host, usually marks the virulence profile of invasive Candida spp. In this paper, we compare virulence of two clinical isolates of *C. albicans* with that of laboratory-induced resistant strains by challenging *G. mellonella* larvae with these pathogens along with monitoring transcriptional profiles of drug efflux pumps genes CDR1, CDR2, MDR1 and the adhesin genes ALS1 and HWP1. Although both clinical isolates were found resistant to both fluconazole and micafungin they were found less virulent than laboratory-induced resistant strains. An unexpected behavior emerged for the former clinical isolate in which three genes, CDR1, CDR2 and HWP1, usually correlated with virulence, although hyper-expressed, conferred a less aggressive phenotype. On the contrary, in the other isolate, we observed a decreased expression of CDR1, CDR2 and HWP1 as well as of MDR1 and ALS1 that may be consistent with the less aggressive performance observed in this strain. These altered gene expressions might directly influence Candida virulence or they might be an epiphenomenon of a vaster rearrangement occurred in these strains during the challenge with the host's environment. An in-deepth comprehension of this scenario could be crucial for developing interventions able to counteract *C. albicans* invasiveness and lethality.

## Introduction

*Candida albicans* is an invasive pathogenic organism causing a broad spectrum of opportunistic infections ranging from mucocutaneous infections to invasive mycoses of deep-seated

https://wwwuniroma1.it The funders had no role in study design, data collection and analysis, decision to publish, or preparation of the manuscript.

**Competing interests:** The authors have declared that no competing interests exist.

organs [1,2]. In nosocomial settings, it is recognized as the main cause of morbidity for patients underwent surgery or implanted with medical devices or treated with long course antibiotic therapy. Bloodstream infections may be life-threatening particularly for those patients whose immune system is suppressed or compromised. In adults, an accurate estimate of the attributable mortality to invasive candidiasis has been recently reported to range from 10% to 20% [2]. Nevertheless, *C. albicans* is also a commensal organism present in more than 60% of healthy individuals, being part of the normal human microbiome, localized in the gastrointestinal tract, over the skin, in the oral cavity and in other mucosal surfaces [2,3]. Transition from commensalism towards a systemic infection requires an increased colonization, generally associated with a local or general failure in host defences [4]. Invasive colonization depends upon the conversion of the yeast to the hyphal form, morphologies that *C. albicans* can assume during its life cycle responding to different environmental conditions [3]. This peculiar feature distinguishes Candida from other fungal species whose life forms are either yeast as *Cryptococcus neoformans* or filamentous hyphal structures as Aspergillus spp. [5]. Transition to hyphal form is an important trait for the virulence of *C. albicans* along with the ability of developing a resistance to antifungal drugs after a prolonged treatment of specific chemotherapy agents [6]. Molecular mechanism responsible for resistance to the two major class of antifungal drugs, azole and echinocandins have been investigated. As far as azole resistance is concerned, it correlates mainly with an increased expression of drug efflux membrane transporters due to upregulation of multidrug transporters belonging to ABC (ATP-binding cassette) transporters, CDR1 and CDR2 or to the major facilitators as MDR1. Overexpression of CDR1, CDR2, and MDR1 is commonly observed in azole-resistant oral, systemic, and vaginal *C. albicans* clinical isolates [7]. Beyond drug resistance, a basic event in Candida infection is adherence to host surfaces, which is required for initial colonization. Adherence contributes to persistence of the organism within the host, and it is thus considered essential for the spreading and the settling of the fungus. Working together, the transition to hyphal form and adherence cause damage to the host mucosa by the combined action of secreted aspartyl proteases and phospholipases thereby facilitating the invasion of the organism into the epithelium [8,9]. In both commensal and pathogenic lifestyles, *C. albicans* utilizes a set of proteins called adhesins to prime adherence among *C. albicans* cells, between C. albicans and host cells or with inanimate surfaces. Adhesins belong to ALS (ALS1-7 and ALS9) and HWP families both comprising GPI-anchored proteins localized on cell wall [10,11]. Among ALS proteins, Als1p has a key role in adhesion being involved in almost all adhesive interaction of *C. albicans* with endothelial and epithelial cells, glycans and abiotic surfaces [12]. Hwp1p, a transglutaminase substrate, plays a role in adherence to epithelial cells but is not required for adherence to endothelial cells [13,14]. A synergic effect for Als1p and Hwp1p has been reported for germ tube formation and development, a fondamental stage for the morbidity of the fungus [15].

Analysis of expression of these virulence genes in *C. albicans* organisms collected from different environments as those isolated from patients with invasive candidiasis, may help to understand the variability of virulence reported for this fungus. As a matter of fact, clinical isolates acquire a more virulent behaviour when compared with reference strains confirming the establishment of more invasive traits in these pathogens [16]. In this paper we compare the virulence and lethality of two clinical isolates with those of three laboratory-induced resistant (LIR) strains to fluconazole or micafungin or both, by challenging *G. mellonella* larvae with these pathogens along with monitoring transcriptional profiles of CDR1, CDR2, MDR, ALS1 and HWP1 genes. This approach may gain insight on the relathiship existing between these molecular factors and virulence in organisms either isolated from human host or developed as coltured strains.

## Materials and methods

### Yeast strains and growth conditions

The strain $CO23_S$ of *C. albicans* was isolated from a subject with vulvo-vaginal candidiasis. It was kindly donated by Prof. Cassone of Istituto Superiore di Sanità in 2001 and since then it has been used in a number of studies [17]. This strain was originally susceptible to micafungin (FK463, MIC 0,025 μg/mL) and fluconazole (FLC MIC 0.25 μg/mL). It was made resistant to FK463 ($CO23_{RFK}$; MIC >4 μg/mL) or to FLC ($CO23_{RFLC}$; MIC >64 μg/mL) as previusly reported [18]. The micafungin resistant strain ($CO23_{RFK}$) was made double resistant, ($CO23_{RR}$), by ten growth passages in stepwise increasing concentrations of fluconazole (0.32 to 128 μg/mL) in agar solidified Yeast Nitrogen Base (YNB) medium, at 28˚C. The double resistant phenotype remained stable after multiple passages in culture. The two clinical isolates, CI1 and CI2, of *C. albicans* included in the present study, were identified as a collection of clinical specimens isolated from blood culture.

### Minimal Inhibitory Concentration (MIC)

The Minimal Inhibitory Concentration (MIC) was determined by micro-broth dilution method according to the Clinical and Laboratory Standards Institute/National Commitee for Clinical Laboratory Standards (CLSI/NCCLS) Approved Standard M27-A3, 2008 [19]. Fluconazole and micafungin 0.5 g/L solutions were prepared by dissolving each drug in endotoxin-free water. Dilutions of fluconazole and micafungin ranging from 0.06 to 64 μg/mL and 0.007 to 8 μg/mL respectively, were prepared in 96-well plates. The inoculum size was about $2.5 \times 10^3$ cells/mL. The plates were incubated at 28˚C for 24–48 h.

### *Galleria mellonella* infection model

Ten larvae of *Galleria mellonella* (250–320 mg each) were randomly selected for each step of the procedure. Larvae were stored at 15˚C before use and starved for 24 h before infection as recommended. $2 \times 10^3$ cells/larvae of each *C.albicans* strains in PBS buffer were injected into the hemocoel through the last left proleg (Hamilton syringe 701N, volume 10 μL, needle size 26 s, cone tip) [20]. After injection, larvae were incubated in petri dishes at 37˚C in standard aerobic conditions and survival was recorded at 24 h intervals for five days. Larvae were considered dead when they displayed no movement in response to gentle prodding with a pipette tip. Samples untreated or injected with PBS were both utilized as controls. Each experiment was repeated at least three times.

### Extraction of RNA and RT-PCR

Total RNA was extracted by RNeasy Mini Kit (Qiagen) from $5 \times 10^7$ cells. Approximately 6 μg of total RNA were used as template to synthesize the cDNA (final volume, 20 μl) by Tetro cDNA Synthesis Kit (Bioline). 1 μl of cDNA was used for amplification reaction by SensiMix SYBR & Fluorescein Kit (Bioline) in a final volume of 25 μl. RT-qPCR conditions consisted of 1 cycle at 95˚C (4 min); 40 cycles of 94˚C (10 s), 64˚C (30 s) followed by a 81 step melting–curve analysis (initial temperature 55˚C, increasing 0.5˚C every 10 seconds). Real time PCR data were acquired and analyzed using SensiMix SYBR & Fluorescein Kit (Bioline). The beta-actin mRNA gene, was used as internal control, and the expression values of CDRs, MDR1, ALS1 and HWP1 were normalized against the ACT1 transcript. The sequences of the RT-PCR primers, including the housekeeping actin gene ACT1, are listed in Table 1. Data were expressed as the mean values ± SEM from at least three independent experiments. A comparative threshold cycle (CT) method was used to analyze the RT-PCR data. Sensitive *Candida*

**Table 1. RT-PCR primers.**

| GENE | Forward Primer (5′–3′) | Reverse Primer (5′–3′) |
|------|----------------------|------------------------|
| CDR1 | AACCGTTTACGTTGAACACGATAT | ACCAACTTCACCATCTTCAATGAC |
| CDR2 | TGGCTAGTGTTTATATGGCAACCT | AAGCTTCAGCAATTGACACTCTTT |
| MDR1 | TCTCGGGTGGATTCTTTGCTAAT | AATGGACCAAAACTAGGACCACA |
| HWP1 | AGGTAGACGGTCAAGGTGAAACAG | TGGCTCTTGTGGTTGTTGTTGTGTG |
| ALS1 | GCAAACCCAGGAGACACATTCAC | AACACCGTCAGCAGTCAAATCAAC |
| ACT1 | GACAAATGGGTAGGGTGGGAAAAC | TGTGACAGTAACATCCCAAACGAG |

*albicans* sample (CO23s) was used as a control and target gene Ct values were normalized against actin. Data were analyzed by using the $2^{-\Delta\Delta CT}$ method and expressed as fold change with respect to the control [21]. Statistical analyses were performed using one-way analysis of variance (ANOVA) with *Tukey's* post hoc *test* comparison (Graphpad Software Inc., USA). Results were considered statistically significant with $p < 0.05$.

## Hyphae development assay

The hyphae development assay started by incubation of a pre-culture in 50 mL of Sabouraud broth at 28˚C for 24 h. Cells were then recovered and suspended in 5 mL of RPMI1640 supplemented with morpholinepropanesulfonic acid (MOPS) and foetal calf serum (10% v/v) to a final concentration of $1.4 \times 10^7$ cells/mL. After 24 h incubation at 37˚C under agitation, an aliquot was taken and photographed with a phase contrast microscope using a 40x objective. (Optica Microscope, Italy)

## Results

### Antimicrobial activity on *C. albicans* strains

*C. albicans* strains analyzed in this study are reported in Table 2. All of them were assayed for antimicrobial activity of fluconazole (FLC) and micafungin (FK) and results are reported in Table 3. MIC values obtained for CO23$_S$, CO23$_{RFK}$ and CO23$_{RFLC}$ strains were in agreement with those previously reported [18]. MIC values obtained for CO23$_{RR}$ strain confirmed the double resistance of this strain for both FLC and FK drugs, yielding values of 32 μg/mL for FLC and 8 μg/mL for FK. Analyses of both clinical isolates 1(CI1) and 2 (CI2) showed a double resistance for FLC and FK with MIC values of CI1 overlapping those of the CO23$_{RR}$ strain. On the contrary, in CI2, an halved MIC value of 16 μg/mL for FLC was obtained compared with CO23$_{RR}$ strain, while the same MIC value of 8 μg/mL for FK was observed.

**Table 2. Analyzed strains.**

| Abbreviation | Strain information |
|--------------|--------------------|
| CO23$_S$ | CO23 sensitive |
| CO23$_{RFK}$ | CO23 micafungin resistant* |
| CO23$_{RFLC}$ | CO23 fluconazole resistant* |
| CO23$_{RR}$ | CO23 micafungin and fluconazole resistant* |
| CI1 | Clinical isolate 1 |
| CI2 | Clinical isolate 2 |

*Laboratory Induced Resistant (LIR).

**Table 3.  Antimicrobial activity (MIC) of Fluconazole and Micafungin on different *C.albicans* strains.**

| Strains | Micafungin μg/mL | Fluconazole μg/mL |
|---|---|---|
| CO23$_S$ | 0.025 | 0.25 |
| CO23$_{RFK}$ | 8 | 0.25 |
| CO23$_{RFLC}$ | 0.25 | 64 |
| CO23$_{RR}$ | 8 | 32 |
| CI1 | 8 | 32 |
| CI2 | 8 | 16 |

## Infection in *G.mellonella* of sensitive and resistant *C.albicans* strains

Laboratory-induced resistant (LIR) strains and clinical isolates were assayed for their pathogenicity using a larvae systemic infection model. Results obtained using this approach, are reported as a Kaplan-Meier survival curve, for each strain, in Fig 1. The wild type CO23$_S$ strain caused an overall mortality of about 40% after 48 h and 50% after 72 h. An increased lethality of 70% was measured for LIR CO23R$_{FLC}$ and CO23$_{RFK}$ after 48 h, whereas 90% of mortality was observed for the double LIR CO23$_{RR}$ strain. Clinical isolates CI1 and CI2, despite their double resistance to FLC and FK, showed a lethality of only 70% similar to that observed for LIR CO23$_{RFLC}$ and CO23$_{RFK}$ strains. However, this value was reached up to a tripled time (120 h instead of 48 h for CI1), with respect to the LIR CO23$_{RFLC}$ and CO23$_{RFK}$ strains. Furthermore, this value was much lower than 90% observed for double LIR strain although CI1 and CI2 shared the same double resistance profile.

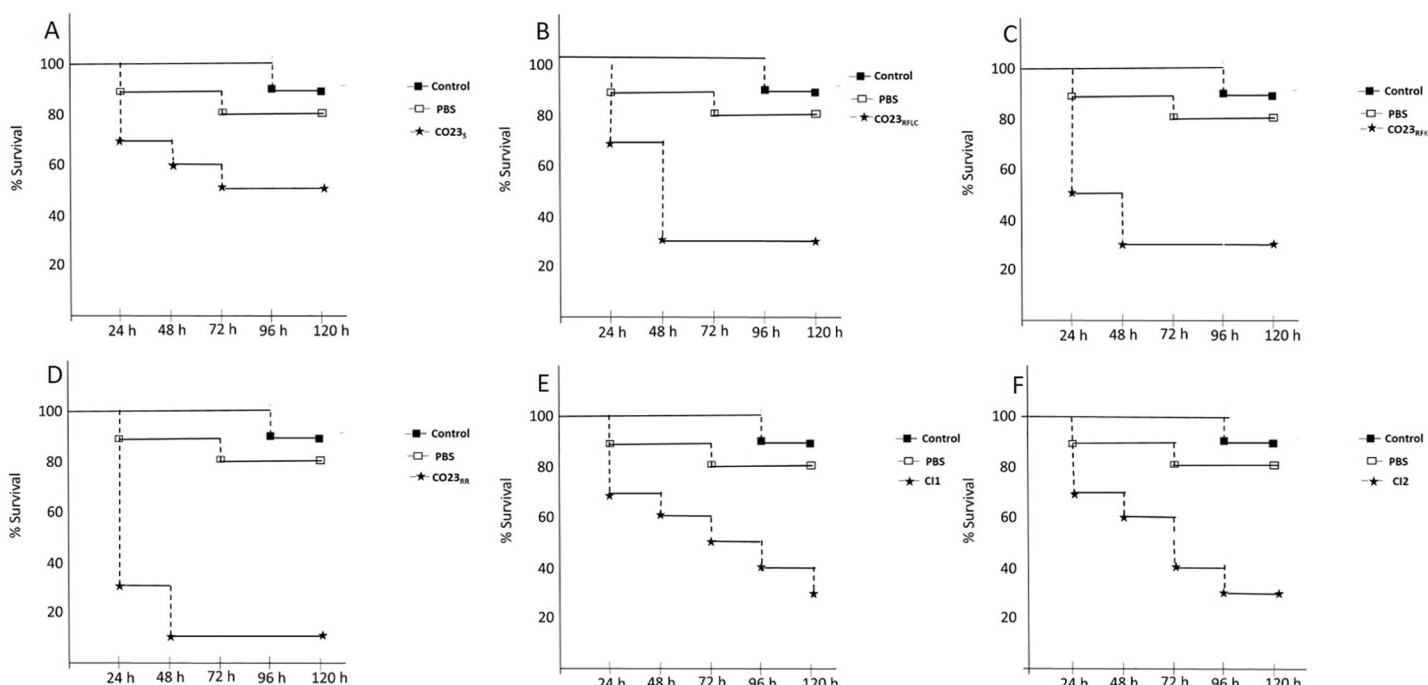

**Fig 1.  Infection in *G.mellonella* by sensitive and resistant *C.albicans* strains.** Comparison of sensitive and resistant strains of *C.albicans* virulence in *G. mellonella* infected with 2x10$^3$ fungal cells/larvae and incubated at 37°C up to 5 days. (A) CO23$_S$,(B) CO23R$_{FLC}$, (C) CO23$_{RFK}$, (D) CO23$_{RR}$, (E) CI1 (F) CI2. Results are expressed as % survival. Experiments with samples untreated or injected with PBS are reported as controls. Each experiment was repeated three times.

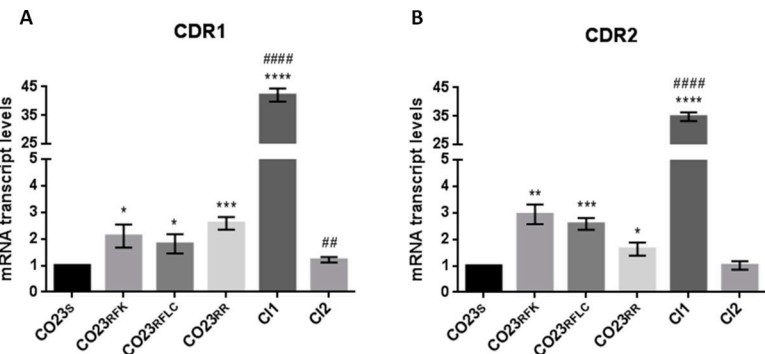

**Fig 2. Expression levels of *C. albicans* CDR1 and CDR2 genes determined by quantitative Real-Time PCR.** The expression level of the CDR1 (A) and CDR2 (B) genes in the wild-type (CO23), resistant (CO23RFK, CO23RFLC and CO23RR) strains and Clinical Isolates (CI1 and CI2) are represented as n-fold increase or decrease relative to the level of the control strain (CO23). Data are shown as mean±SEM from three independent experiments performed in triplicate. $^*p< 0.05$, $^{**}p< 0.01$, $^{***}p< 0.001$, $^{****}p<0.0001$ *vs* CO23$_S$ and $^\#p< 0.05$, $^{\#\#}p< 0.01$, $^{\#\#\#}p< 0.001$, $^{\#\#\#\#}p< 0.0001$ *vs* CO23$_{RR}$.

## Molecular analyses of drug resistance-related genes in Candida strains

The unexpected behaviour observed for CI1 and CI2 in the *G. mellonella* assay asked for an in-depth investigation of genes involved in *C. albicans* virulence. To this pourpose, we examined genes frequently associated with increased fungal pathogenic profiles as those codifying for multidrug efflux pumps involved in drug resistance or adhesins, cell wall proteins necessary for fungal colonization. CDR1, CDR2 and MDR1 were selected for the former category while ALS1 and HWP1 for the latter. mRNA expression levels of these genes were normalized against transcripts of the housekeeping ACT1 gene, and are reported in Figs 2–4 as a ratio with the sensitive strain CO23$_S$. As expected from our previous work [18], CDR1 and CDR2 genes were overexpressed in LIR CO23$_{RFK}$, CO23$_{RFLC}$ and CO23$_{RR}$ strains compared to CO23$_S$. Clinical isolated showed a markedly different expression of these genes: CI1 overexpressed CDR1 and CDR2 in an astonishing fashion while CI2 showed only a small increase with respect to CO23$_S$ strain (Fig 2). MDR1 was overexpressed in all LIR strains while a lower expression was recorded in CI1 and particularly in CI2 where its value was comparable with that of the CO23$_S$ strain (Fig 3).

The second set of genes we selected, ALS1 and HWP1, belonging to the adhesin families, are considered as the principal virulence factors in Candida pathogenesis, for enabling the fungus to tightly adhere to host tissues. Fig 4 (panel A) shows the histogram of the mRNA expression level of ALS1 with values expressed as a ratio with the sensitive CO23s strain. All LIR strains showed a moderate increase in their ALS1 expression along with the CI2 strains while a threefold increase was observed for the CI1. As far as expression of HWP1 is concerned, only CI1 expression deviated from the other resistant strains including the CI2 strain. All of them showed an HWP1 expression similar to that observed for the sensitive CO23s strain while CI1 expression was fivefold (Fig 4 panel B).

## Hypha formation

As shown in Fig 5, in the hypha development assay, all strains, sensitive and resistant, were able to produce hyphal filaments when incubated in an enriched serum medium after 24 h incubation. However, hyphae produced by CI1 were clearly more numerous and longer than those generated by the other strains. This result confirmed that the increased expression of HWP1 gene in strain CI1 defined a phenotype more prone to develop a more complex hyphal form.

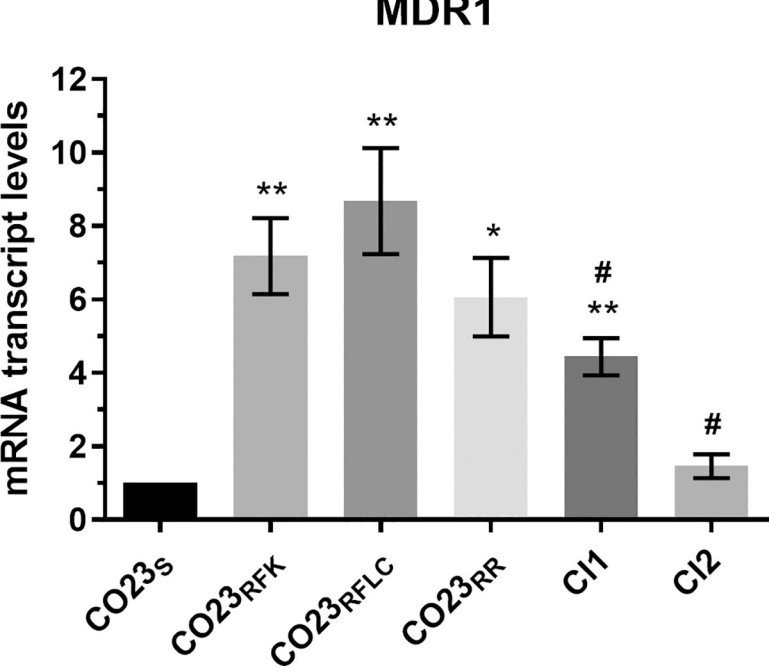

**Fig 3. Expression levels of *C. albicans* MDR1 gene determined by real-time PCR.** The expression level of the MDR1 gene in the wild-type (CO23), resistant (CO23RFK, CO23RFLC and CO23RR) strains and Clinical Isolates (CI1 and CI2) are represented as n-fold increase or decrease relative to the level of the control strain (CO23). Data are shown as mean±SEM from three independent experiments performed in triplicate. $^*p < 0,05$, $^{**}p < 0,01$, $^{***}p < 0.001$, $^{****}p < 0,0001$ *vs* CO23$_S$ and $^\#p < 0.05$, $^{\#\#}p < 0,01$, $^{\#\#\#}p < 0.001$, $^{\#\#\#\#}p < 0,0001$ *vs* CO23$_{RR}$.

## Discussion

Candida is part of the microbiome of healthy people and do not manifest pathological features thanks to the barrier of the immune system against the spread of the fungus in the body. In immunocompromised patients, this barrier is broken and therapeutic intervention is required to contain Candida spreading. Major drugs used for this purpose, as those belonging to azoles and echinocandins families, have mainly a static effect on the fungal growth. Their action

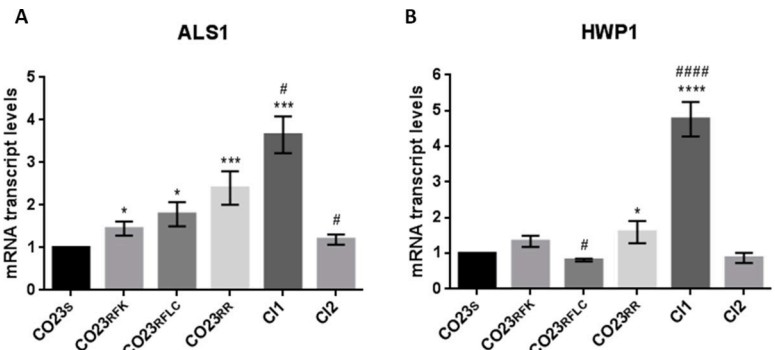

**Fig 4. Expression levels of *C. albicans* ALS1 and HWP1 genes determined by real-time PCR.** The expression level of the ALS1 (A) and HWP1 (B) genes in the wild-type (CO23), resistant (CO23RFK, CO23RFLC and CO23RR) strains and Clinical Isolates (CI1 and CI2) are represented as n-fold increase or decrease relative to the level of the control strain (CO23). Data are shown as mean±SEM from three independent experiments performed in triplicate. $^*p < 0.05$, $^{**}p < 0.01$, $^{***}p < 0.001$, $^{****}p < 0.0001$ *vs* CO23$_S$ and $^\#p < 0.05$, $^{\#\#}p < 0,01$, $^{\#\#\#}p < 0.001$, $^{\#\#\#\#}p < 0,0001$ *vs* CO23$_{RR}$.

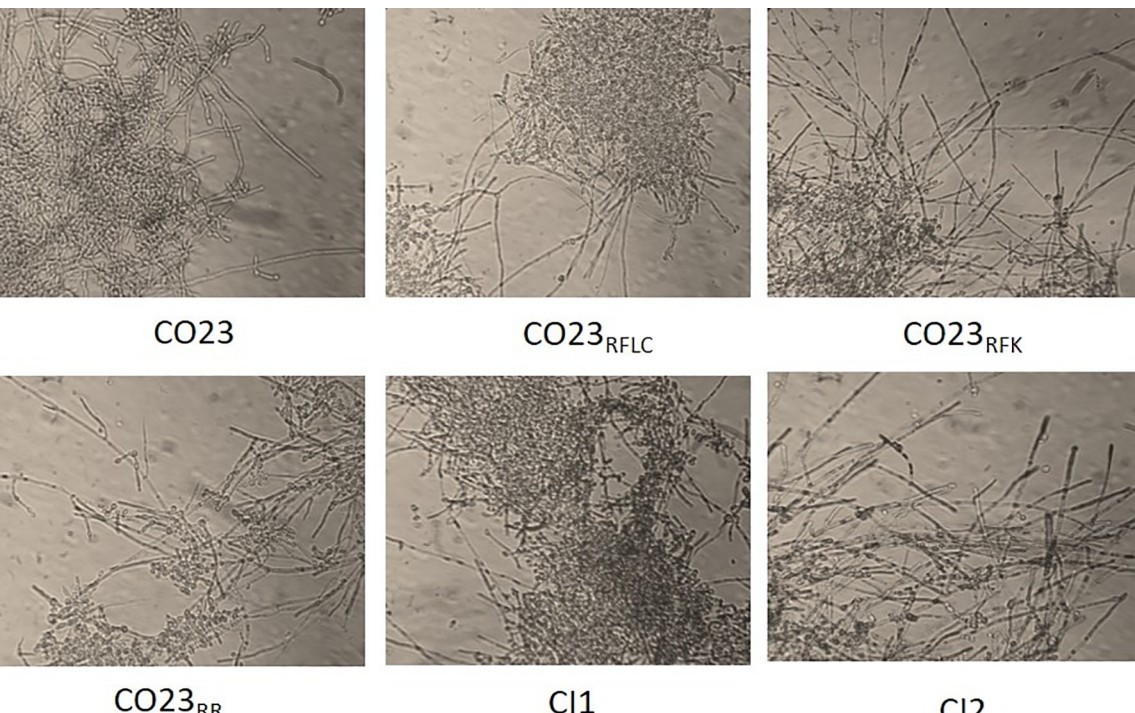

**Fig 5. Filamentation in *C.albicans*.** Hyphal growth of *C. albicans* strains CO23$_S$, CO23$_{RFK}$, CO23$_{RFLC}$, CO23$_{RR}$, CI1 and CI2. Cells were induced to form hyphae by 24 h incubation at 37˚C in RPMI 1640 medium containing 10% fetal calf medium. Photomicrographs were taken by a phase contrast microscope using a 40x objective and they are representative of 50% random fields observed.

limits the fungal multiplication although, not killing the fungus, they give it a chance to develop a drugs resistance [6].

In this paper we analyzed two clinical isolated both resistant to fluconazole and micafungin in comparison with laboratory induced-resistance (LIR) strains in order to assess the host's influence on modulating molecular and biological features of these clinical resistant strains.

Assayes on *Galleria mellonella*, a widely used host for systemic infection model, showed a marked difference between clinical isolates and LIR strains. The formers were less aggressive than LIRs especially when compared with the double LIR strain. This finding was very peculiar considering that both clinical isolates resulted resistant to both micafungin and fluconazole even if CI2 had an halved value for fluconazole compared with double LIR strain. To investigate the molecular bases of the different degree of virulence observed between laboratory and clinical strains, we focused our investigation on the expression of genes whose association with Candida virulence has been well established as those involved in drug export and adhesins. We selected *Candida* drug resistance (CDR) genes, and in particular CDR1 and CDR2 encoding for multidrug pumps responsible for the export of fluconazole. RT-PCR analyses showed an increase of both CDR1 and CDR2 transcription in all resistant strains but, in CI1, transcription level was dramatically enhanced (more than 30-folds) for both genes. This overexpression, well beyond the level observed in the other resistant strains, indicated that a disfunction in the regulation of these genes was likely to occur. Regulation of CDR1 and CDR2 genes is a complex pathway whose map is still under construction. CDR1 promoter presents multiple regulators elements such as AP1, heat shock, drug response and steroid responsive elements. Both upregulating and downregulating transcription factors have been described so far and among them the Transcriptional activator of CDR genes (Tac1p), Ncb2, the β subunit of NC2

complex, the Uptake Control 2 (Upc2) and Cap1, were shown to be involved in the activated transcription of CDR1 gene in azole resistant isolate (AR) of *C. albicans* [22,23]. In particular, Tac1p was shown to control expression of both CDR1 and CDR2 and mutation of this protein has been demonstrated to be responsible for hyperactivation of alleles expressing CDRs [24]. It has been recently reported that hyperactivation resides on a gain-of-function mutation on a single nucleotide N977D of TAC1 changing the properties of the C-terminal region of the protein able to enhance CDR1 and CDR2 expression in a still unclarified manner [25]. Since the mutated allele is codominant with the wild type allele, high level of expression is likely to be obtained in presence of homozygosity of the mutated TAC1 [26]. A similar scenario might explain the considerable overexpression of both CDR1 and CDR2 observed for the CI1. However, MIC value observed for CI1, similar to those reported for the LIR strains, clearly indicated that the enhanced transcription did not correlate with an increased resistance to the antifungal drugs suggesting either that CDRs RNAs were not translated or that other factors might interfere with the CDRs ability to export the drugs.

A comprehensive estimation of the efflux capacity towards the antifungal drugs from Candida cell should consider not only CDRs but also MDR1 pump belonging to the major facilitator superfamily selective for fluconazole. Expression of this drug efflux pump is under control of the multidrug resistant regulator 1 (MRR1) and its costitutive activity may reside on gain-of-function-mutations occurring in clinical isolates from human patients after prolonged fluconazole treatment [27]. RT-PCR of MDR1 showed similar values for all LIR and CI1 strains but not for CI2 where this gene was expressed at a level similar to the CO23 wild-type strain probably indicating an impairment of MRR1 action. These data are in agreement with the MIC value for fluconazole and can explain the halved MIC of CI2 with respect to the values of both CO23RR and CI1. We than extended our analysis to ALS1 and HWP1 genes, deeply involved in the virulent behaviour of the fungus. Both proteins encoded by these genes belong to larger families formed by many members and collectively named adhesins. These proteins are localized on the external face of the cell wall and, through different mechanisms, enable the fungus to adhere tightly to host tissues and therefore are considered important virulence factors for Candida invasivness. RT-PCR analyses of ALS1 and HWP1 showed that in CI1, both ALS1 and HWP1 expression were increased with respect to both double LIR and CI2. In the case of ALS1, increase was 1,5 fold with respect to values observed for double LIR and CI2 while a fivefold increased value was detected for the HWP1 gene. This overexpression along with that observed for CDRs, further pointed out altered pathways of gene regulation in this isolate. However, regulation pathways of this gene, differs either from regulon analyzed above for CDRs gene or from that of ALS1. HWP1 regulation mainly resides on the activity of two transcription factors: Efg1p and Nrg1p. Both factors recognize a 368 bp region localized inside the 2kb promoter termed HWP1 Control Region (HCR). On this site, subregions responsible for activation or repression of HWP1 transcripsion and sensitive to the action of Efg1p and Nrg1p have been mapped [28]. Instead, ALS1 expression is under control of a cAMP activated PKA coded by TPK1 and TPK2 expressing two catalytic subunits and by BCY1 encoding the regulatory portion of the kinase. PKA activation is under control of the conserved RAS/cAMP/PKA pathway that in fungi, operates in a variety of essential cellular metabolisms such as morphological transitions, sexual reproduction and nutrient sensing [29]. In CI1, the huge overexpression of HWP1 together with the significative increase of ALS1 expression may be the cause of the remarkable hyphal development observed for this strain in the filamentation assay when compared with the other analyzed strains [15]. However, it should be noted that the filamentation process involves a complex of interaction based on expression of different genes such as, for instance, QDR1, QDR2 and QDR3, the quinidine drug resistant family proteins belonging to the mayor facilitator superfamily [30]. Although ALS1 and HWP1 are

considered reliable markers of Candida virulence and are overexpressed in the pathogenic organism causing several diseases (i.e vulvovaginitis, periodontal diseases), an increase of their expression in CI1 did not manifest itself as an increase in virulence and aggressivity of the fungus in the *G. mellonella* assay, indicating that they are necessary for a virulent behavior of the parasite but in these cases their overexpression is not positively correlated with lethality.

Taking data on gene expression together, an unexpected behavior emerged for the CI1in which three genes, CDR1, CDR2 and HWP1, usually correlated with virulence, although overexpressed, conferred a less aggressive phenotype. To our knowledge, this is the first report that links virulence genes hyperexpression to such a decrease in lethality. On the contrary, for CI2, decrease in expression of CDR1, CDR2 and HWP1as well as of MDR1 and ALS1 may be consistent with the less aggressive performance observed in this strain. Altered gene expression we observed in clinical isolates might also be just an epiphenomenon of a vaster rearrangement occurring in these strains during the challenge with the host environment. Both ALS1 and HWP1 gene products are cell wall proteins able to elicit a host immune response which in turn may trigger a wide change at molecular level. As these rearrangements may lessen the lethality of the fungus as we observed in our clinical isolates, it will be worthwhile to deepen our knowledge with a more extensive investigation of the main regulatory pathways and intracellular responses that play a significant role in Candida virulence and lethality [31]. However, from this work, it should be clear that resistant clinical isolated are not the actual identity of the laboratory strains and so direct translation of information obtained from laboratory strains should be consider with caution.

## Author Contributions

**Conceptualization:** Bruno Maras, Giuseppina Mignogna, Letizia Angiolella.

**Data curation:** Bruno Maras, Anna Maggiore, Giuseppina Mignogna, Maria D'Erme.

**Formal analysis:** Anna Maggiore, Giuseppina Mignogna.

**Funding acquisition:** Bruno Maras, Giuseppina Mignogna, Letizia Angiolella.

**Methodology:** Anna Maggiore, Giuseppina Mignogna, Letizia Angiolella.

**Supervision:** Bruno Maras, Giuseppina Mignogna, Maria D'Erme.

**Validation:** Bruno Maras.

**Writing – original draft:** Bruno Maras, Giuseppina Mignogna, Letizia Angiolella.

**Writing – review & editing:** Bruno Maras, Giuseppina Mignogna.

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
