## [Decision Letter · Decision Letter 0]

22 Mar 2021

PONE-D-21-05206

Hyperexpression of CDRs and HWP1 genes negatively impacts on Candida albicans virulence

PLOS ONE

Dear Dr. Angiolella,

Thank you for submitting your manuscript to PLOS ONE. After careful consideration, we feel that it has merit but does not fully meet PLOS ONE’s publication criteria as it currently stands. Therefore, we invite you to submit a revised version of the manuscript that addresses the points raised during the review process.

We look forward to receiving your revised manuscript.

Kind regards,

Aijaz Ahmad, Ph.D.

Academic Editor

PLOS ONE

Journal Requirements:

'The authors thank “Cooperativa Comunale Logistica Industriale Integrata” for the donation to support research.'

'LA, GM, BM. Grant number: RM 1181641E9150B3 Sapienza University of Rome https://wwwuniroma1.it

The funders had no role in study design, data collection and analysis, decision to publish, or preparation of the manuscript.'

b. Additionally, because some of your funding information pertains to commercial funding, we ask you to provide an updated Competing Interests statement, declaring all sources of commercial funding.

In your Competing Interests statement, please confirm that your commercial funding does not alter your adherence to PLOS ONE Editorial policies and criteria by including the following statement: "This does not alter our adherence to PLOS ONE policies on sharing data and materials.” as detailed online in our guide for authors  http://journals.plos.org/plosone/s/competing-interests.  If this statement is not true and your adherence to PLOS policies on sharing data and materials is altered, please explain how.

c. Please include the updated Competing Interests Statement and Funding Statement in your cover letter. We will change the online submission form on your behalf.

Reviewers' comments:

Reviewer's Responses to Questions

**Comments to the Author**

1. Is the manuscript technically sound, and do the data support the conclusions?

Reviewer #1: Yes

Reviewer #2: Partly

2. Has the statistical analysis been performed appropriately and rigorously? 

Reviewer #1: Yes

Reviewer #2: Yes

3. Have the authors made all data underlying the findings in their manuscript fully available?

Reviewer #1: Yes

Reviewer #2: Yes

4. Is the manuscript presented in an intelligible fashion and written in standard English?

Reviewer #1: Yes

Reviewer #2: No

5. Review Comments to the Author

Reviewer #1: 1) Line no. 20,21,176,234, 240: There are either typing error or some spelling mistakes in words like microbiome, population, micafungin etc. Please modify.

2- Why the author used different temperature conditions for the solid (Line-98) and liquid media (Line-108) for adapting and calculating the MICs for the strains. Please explain the rationale behind this.

3-What is status of other echinocandin drugs like Caspofungin, Anidulafungin on these clinical isolates which showing resistant to micafungin.

4- Cdr1 is known to be promiscuous transporters involve in transport of various substrates drug specially azoles however there is no report for Cdr1 as transporter for Micafungin. What are the possible reasons for the upregulation of CDR1 in CO23RFK which are resistant to Micafungin and sensitive to Fluconazole?

5- What is the status of expression profile of other CDRs like Cdr4 and Cdr3 in these strains taken under study?

6- Has the author tried gene other than ACT1 as internal control since the CT value may vary from strains to strains also there is no mentioning of primer validation was done before final experiments?

7- Author did not mention the which method (for example Livak method etc.) used for the calculation of relative expression profiling of different gene.

8- Author mentioned the role of MFS transporters specifically MDR1 however the MFS is family containing large numbers of proteins so it’s not possible to check the expression profile for all. Some of the known transporters like QDR (quinidine drug resistance) family of genes encodes transporters belonging to the MFS (major facilitator superfamily) of proteins need to be mentioned or investigate since the QDR genes, individually or collectively, led to defects in biofilm architecture and thickness. So, it can provide better insight of MFS superfamily in these strains.

9- In discussion section author mentioned the role of gain of function mutation in different Tfs have you observed any change after the adaptation in LIR strains also the change in Erg11 sequence or any mutation analysis done?

Reviewer #2: The authors have tried to correlate the virulence of Candida albicans with overexpression of genes related to drug efflux and cell surface adhesion.

-The data does not support the conclusion

- Discussion is majorly speculative

- More studies can be done using other virulence related genes

- Should increase the number of clinical strains and resistant strains

- The choice of abbreviations for the strains (resistant and susceptible) makes the understanding this work difficult

-Tables and figures should have information regarding each strain as footnote

- There are too many typo/spelling mistakes

- Sentence formation is not appropriate at several places throughout the manuscript

6. PLOS authors have the option to publish the peer review history of their article (what does this mean?). If published, this will include your full peer review and any attached files.

Reviewer #1: **Yes: **Dr. Hammad Alam

Reviewer #2: **Yes: **Dr Nikhat Manzoor

---

## [Author Response · Author response to Decision Letter 0]

7 May 2021

General points.

i) English has been revised as well as mis-typings all over the text.

Referee 1

Point 1 Mis-typing: See general point.

Point 2 Temperature condition: we apologize for the mistake. Temperature conditions of the solid media for adapting and the liquid media for MIC were both set at 28° C. Corresponding sentence in Material and Methods, at line 109, has been amended, accordingly.

Point 3 Echinocandin status: the CI1 and CI2 isolates were assayed for caspofungin and MIC value was 8 �g/mL for both isolates, overlapping the results obtained for micafungin.

Point 4 CDR1 upregulation in RFK: we agree that the increase of CDR1 and CDR2 in micafungin resistant strain is not explained with the efflux mechanism usually involved in fluconazole resistant. However, an increase in CDR1 and 2 has been reported as associated with a caspofungin resistance in Candida albicans clinical isolates (Schuetzer-Muehlbauer M, Willinger B, Krapf G, Enzinger S, Presterl E, Kuchler K. The Candida albicans Cdr2p ATP-binding cassette (ABC) transporter confers resistance to caspofungin. Mol Microbiol. 2003 Apr;48(1):225-35. doi: 10.1046/j.1365-2958.2003.03430.x). To confirm CDRs role, authors expressed these genes in Saccharomyces cerevisiae and observed an increased MIC for caspofungin. It is worth notice that the authors stated that resistance is probably not due to the direct effect of these proteins as efflux pumps but more reasonably: “as a consequence of distinct mechanisms that may operate simultaneously”. A similar scenario may be tentatively proposed for our micafungin resistant strain.

Point 5 CDR3 and CDR4: we missed the analyses for these two drug transporters because, to our knowledge, they seem not to have a critical role in resistance to fluconazole. CDR3 was found highly expressed in WO-1 opaque cells and its overexpression was not linked to an increase in fluconazole resistance. This data suggested a role for this protein in the white-opaque switch rather than in drug efflux. (Balan I, Alarco AM, Raymond M. The Candida albicans CDR3 gene codes for an opaque-phase ABC transporter. J Bacteriol. 1997 Dec;179(23):7210-8. doi: 10.1128/jb.179.23.7210-7218.1997.). A similar consideration may be proposed for CDR4. CDR4 mutant revealed not hypersusceptible to fluconazole and furthermore Candida isolates from AIDS patients treated with fluconazole did not reveal any involvement of CDR4 in resistance (Franz R, Michel S, Morschhäuser J. A fourth gene from the Candida albicans CDR family of ABC transporters. Gene. 1998 Oct 5;220(1-2):91-8. doi: 10.1016/s0378-1119(98)00412-0.)

Point 6 RT-PCR internal control: We tried to assess the expression rate using 18S rRNA as housekeeping gene for CDR1 and CDR2 (unfortunately, we had not a suitable amount left for screening the other genes). Analyses showed a similar pattern obtained with the 18S rRNA to that reported in the paper obtained with actin gene. Furthermore, these results agree with those reported in a previous paper where TEF3 gene was used as housekeeping (Angiolella L, Stringaro AR, De Bernardis F, Posteraro B, Bonito M, Toccacieli L, Torosantucci A, Colone M, Sanguinetti M, Cassone A, Palamara AT. Increase of virulence and its phenotypic traits in drug-resistant strains of Candida albicans. Antimicrob Agents Chemother. 2008 Mar;52(3):927-36. doi: 10.1128/AAC.01223-07). Data are reported in the figures below. (Please note that only two replicates for CDR2 were performed due to the small availability of this sample and thus statistical analysis was missed). 

Expression levels of C. albicans CDR1 and CDR2 genes determined by quantitative Real-Time PCR.

The expression levels of the CDR1 and CDR2 with Actin as housekeeping (A and C) and CDR1 and CDR2 with 18S rRNA as houskeeping (B and D) in the wild-type (CO23), resistant (CO23RFK, CO23RFLC and CO23RR) and clinical isolated (CI1 and CI2) strains, represented as n-fold increase or decrease relative to the level of the control strain (CO23). Data for CDR1s and CDR2 with Actin housekeeping and for CDR1 with 18S rRNA are shown as mean±SEM from three independent experiments performed in triplicate. *p< 0.05, **p< 0.01, ***p< 0.001, ****p<0.0001 vs CO23S and #p< 0.05, ##p< 0.01, ###p< 0.001, ####p< 0.0001 vs CO23RR.

Point 7 Calculation of relative expression profiles: A comparative threshold cycle (CT) method was used to analyze the RT-PCR data. Sensitive Candida albicans sample (CO23s) was used as a control and target gene Ct values were normalized against actin. Data were analyzed by using the 2-ΔΔCT method and expressed as fold change with respect to control (Livak KJ, Schmittgen TD. Analysis of relative gene expression data using real-time quantitative PCR and the 2(-Delta Delta C(T)) Method. Methods. 2001 Dec;25(4):402-8. doi: 10.1006/meth.2001.1262.). This sentence has been added in Material and Method in section: Extraction of RNA and RT-PCR on lines 131-134 

Point 8 MSF-QDR: a sentence suggesting an involvement of QDR family in the remarkable hypha development observed in CI1 has been added in the Discussion, lines 311-314 along with the reference (Shah AH, Singh A, Dhamgaye S, Chauhan N, Vandeputte P, Suneetha KJ, Kaur R, Mukherjee PK, Chandra J, Ghannoum MA, Sanglard D, Goswami SK, Prasad R. Novel role of a family of major facilitator transporters in biofilm development and virulence of Candida albicans. Biochem J. 2014 Jun 1;460(2):223-35. doi: 10.1042/BJ20140010.)

).

Point 9 Mutations: genetic analysis of fluconazole resistant strains was missing. We analysed RT-PCR expression of ERG11 in sensitive, flucanozole and micafungin resistant strains and we found no difference in gene expression between these strains (Angiolella L, Stringaro AR, De Bernardis F, Posteraro B, Bonito M, Toccacieli L, Torosantucci A, Colone M, Sanguinetti M, Cassone A, Palamara AT. Increase of virulence and its phenotypic traits in drug-resistant strains of Candida albicans. Antimicrob Agents Chemother. 2008 Mar;52(3):927-36. doi: 10.1128/AAC.01223-07.). In the same paper, we reported a S645Y mutation on FSK1 gene in micafungin resistant strain. 

Referee 2

Point 1 Data do not support conclusion Overexpression of CDRs and MDR is a usual event occurring in fluconazole resistant strains. The increased expression of this efflux pumps is usually correlated with an increased virulence of the fungus. In this paper we found an evidence, the hyperexpression of these genes and HWP1, that opposes to this common view. For this reason, we believe that this report may be a worthwhile information in this research field.

Point 2 Discussion is speculative Unfortunately, data available in literature on regulatory pathways responsible for altered gene expression and reported in the Discussion, do not allow to find a common link in the regulation of the hyperexpression of the three analysed genes. Therefore we refrained from fully correlate reduction of Candida lethality to the altered gene expression rather suggesting that it may be just an epiphenomenon of a vaster rearrangement occurring in these strains.

Point 3 Virulence genes We agree that the panel of virulence genes could be increased. However, focusing on gene expression, CDRs and MDR are frequently overexpressed in fluconazole resistant strains along with ALS1 and HWP1 as far as filamentation and adhesiveness is concerned.

Point 4 Number of clinical strains We believe that the clinical isolate with the hyperexpression of CDR1, CDR2 and HWP1, found by serendipity, represents a very rare event and then it is unlikely to fish again a resistant clinical strains with a similar phenotype even increasing the number of clinical strains analysed. 

Point 5-6: Abbreviations -Tables and figures We agree that it is difficult to memorize all strain abbreviations and therefore we decided to insert a new Table (Table 2) in the Result section, reporting information and abbreviation for each strain.

Point 7-8 English: see general points.

Funding information: we delete in the Acknowledgements section thanks to “Cooperativa Comunale Logistica Industriale Integrata” without sending the founding statement and therefore competing interest statement remain the same as previously.

Founders’ roles have been updated.

---

## [Editor Report · Decision Letter 1]

18 May 2021

Hyperexpression of CDRs and HWP1 genes negatively impacts on Candida albicans virulence

PONE-D-21-05206R1

Dear Letizia Angiolella

We’re pleased to inform you that your manuscript has been judged scientifically suitable for publication and will be formally accepted for publication once it meets all outstanding technical requirements.

Kind regards,

Aijaz Ahmad, Ph.D.

Academic Editor

PLOS ONE

---

## [Editor Report · Acceptance letter]

21 May 2021

PONE-D-21-05206R1 

Hyperexpression of CDRs and HWP1 genes negatively impacts on *Candida albicans* virulence 

Dear Dr. Angiolella:

I'm pleased to inform you that your manuscript has been deemed suitable for publication in PLOS ONE. Congratulations! Your manuscript is now with our production department. 

Kind regards, 

on behalf of

Dr. Aijaz Ahmad 

Academic Editor

PLOS ONE